# Teaching Deprescribing and Combating Polypharmacy in the Pharmacy Curriculum: Educational Recommendations from Thematic Analysis of Focus Groups

**Devin Scott** [1], **Alina Cernasev** [2,*], **Rachel E. Barenie** [3], **Sydney P. Springer** [4] **and David R. Axon** [5]

1 Teaching and Learning Center, University of Tennessee Health Science Center, 920 Madison, Suite 424, Memphis, TN 38163, USA

2 Department of Clinical Pharmacy and Translational Science, University of Tennessee Health Science Center College of Pharmacy, 301 S. Perimeter Park Dr., Suite 220, Nashville, TN 37211, USA

3 College of Pharmacy, University of Tennessee Health Science Center, Memphis, TN 38163, USA

4 Department of Pharmacy Practice, University of New England School of Pharmacy, Westbrook College of Health Professions, 716 Stevens Ave, Portland, ME 04013, USA

5 Department of Pharmacy Practice & Science, University of Arizona College of Pharmacy, 1295 N Martin Ave, Tucson, AZ 85721, USA

* Correspondence: acernase@uthsc.edu

**Abstract:** In the last two decades in the United States (US), the previous research has focused on medication optimization, including polypharmacy. Polypharmacy is associated with several negative outcomes, which may be resolved by deprescribing medications that are no longer necessary. Although deprescribing is a critical aspect of a pharmacist's role, some studies have demonstrated that student pharmacists are less familiar with their future role in deprescribing. Thus, this study aimed to explore student pharmacists' perceptions of deprescribing in the pharmacy curriculum. This qualitative study was conducted with student pharmacists enrolled in three Doctor of Pharmacy (Pharm.D.) programs in the US. The participants, all student pharmacists at the time of the study, were identified via an email requesting their voluntary participation in a focus group study. The focus groups were conducted via an online platform over three months in 2022, and recruitment continued until thematic saturation was obtained. Using thematic analysis, the corpus of the transcribed data was imported into Dedoose®, a qualitative software that facilitated the analysis. Three themes emerged from the data: (1) the importance of deprescribing; (2) barriers to deprescribing; (3) education recommendations. The data highlight that the student pharmacists believe integrating deprescribing content into the clinical, didactic, and simulation education would help them overcome the identified obstacles. Colleges of pharmacy should consider emphasizing the importance of deprescribing in their curriculum, creating programs to assist future pharmacists in addressing the barriers to deprescribing, and adopting the suggested educational strategies to improve the deprescribing education that is offered.

**Keywords:** student pharmacist; US clinical pharmacist; deprescribing

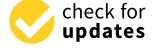



## 1. Introduction

Student pharmacists in the United States (US) study evidence-based medicine and the safety and effectiveness of medications to optimize therapy for their patients [1]. Medication optimization is increasingly important given the aging health status of the US population. For example, in 2012, approximately 60 million people in the US had two or more chronic health conditions that lead to increased morbidity [2]. Increased morbidity leads to an increase in the number of medications being used, as highlighted by a retrospective database analysis that showed 36.8% of individuals were taking more than five medications (i.e., polypharmacy) to manage their chronic conditions between 2009 and

2016 [3]. Polypharmacy is associated with several negative outcomes, which may be mitigated with appropriate deprescribing efforts [4]. Although deprescribing is becoming an important role for pharmacists, many student pharmacists are unfamiliar with the concept of deprescribing and their role in it [5].

Deprescribing has been defined as "the process of withdrawal of an inappropriate medication, supervised by a health care professional with the goal of managing polypharmacy and improving outcomes" [6]. Previous studies have shown that the teaching of deprescribing in professional programs varies [7] and that student pharmacists are often unprepared to recommend or implement deprescribing in practice [8–10]. Another study found that student pharmacists felt more prepared to identify inappropriate medications than medical students [9]. Although symposiums have been held in recent years to discuss deprescribing education [11,12], student pharmacists still need opportunities in their professional curriculum to become equipped for deprescribing activities upon entering clinical practice.

While clinical tools to assist with deprescribing efforts are available, challenges implementing these tools exist. For example, one of the tools focuses on certain drug classes instead of a comprehensive guide for all medications [6]. There is also the precedent of expanding healthcare professionals' roles to formally include deprescribing. For instance, in the nursing community, there have been recent efforts, with a degree of success, to implement aspects of deprescribing into nurses' practice [13]. If deprescribing is to be successful, however, new pharmacists require support from experienced healthcare colleagues [14]. Despite this progress, there is still a lack of knowledge regarding how student pharmacists perceive deprescribing. Thus, this study aimed to explore student pharmacists' perceptions of the implementation of the place of deprescribing in the pharmacy curriculum.

## 2. Methods

*Subjects, Recruitment, Data Collection, and Analysis*

This qualitative study was conducted with student pharmacists enrolled in three different Doctor of Pharmacy (Pharm.D.) programs, including the University of Tennessee Health Science Center (UTHSC, IRB# 21-08234-XM), University of Arizona (IRB# 2021-015-PHPR), and University of New England (UNE, IRB# 0821). The subjects, who were all student pharmacists at the time of the study, were identified via an email that requested their voluntary participation in a focus group (FG) study. Focus groups enhance the potential for producing richer data and obtaining a collective opinion about the research question, because they facilitate brainstorming and discussion between four or more participants simultaneously [15]. Research team members with experience in qualitative data collection (D.S. and A.C.) led all focus groups. Focus groups were conducted via an online platform (Zoom), which enabled students from all three universities to attend [16]. The recruitment occurred simultaneously over three months in the fall semester of 2022 in all three colleges of pharmacy. Interested student pharmacists signed up on a list, and A.C. contacted them to find out the preferred date and time they would like to participate in the study.

A semi-structured focus group strategy was used to allow the students to discuss their experiences with deprescribing in the pharmacy curriculum and during their Advanced Pharmacy Practice Experience (APPE) and Institutional Pharmacy Practice Experience (IPPE) [15]. This study used the theory of planned behavior to conceptualize the semi-structured focus groups [17]. Springer et al., 2022 provided additional information regarding the conceptualization of this study and how the theory of planned behavior guided the development of the questions [8]. The semi-structured strategy allowed the researchers to largely pose the same questions to each group while allowing for additional questions that were raised by earlier discussions to be included in focus groups conducted later. This strategy to incorporate additional questions enhanced the external validity of the study findings [15]. The questions were divided into three topics, which focused on deprescribing education [8].

The virtual focus group transcripts were audio recorded, professionally transcribed, and analyzed using thematic analysis that followed the six-step process as outlined by

Braun and Clarke [18]. The steps included (1) familiarization with the corpus of data; (2) inductively coding the entire dataset; (3) identifying emerging themes; (4) reviewing themes with the research team; (5) defining and naming the themes; and (5) writing the analysis [18]. The analysis team (A.C., D.S., and D.R.A.) continued recruiting subjects until thematic saturation was achieved, at which point no new themes emerged with subsequent focus groups [15]. All focus group transcripts were uploaded to a qualitative analysis software, Dedoose® (Manhattan Beach, CA, USA), which was used for generating initial codes and developing and reviewing themes. Standards for Reporting Qualitative Research (SPQR) criteria for demonstrating the quality of qualitative research were met [19]. A previous manuscript described the methodology in more detail [8].

## 3. Results

Student pharmacists from three colleges of pharmacy in the US were invited to participate in this study. The total number of eligible student pharmacists was 1366 (UNE, N = 158; University of Arizona, N = 526; UTHSC, N = 682). Of these, 26 student pharmacists consented to participate in four focus group discussions. Most participants (N = 16) were enrolled in their fourth year, seven were enrolled in their third year, and three were enrolled in their second year. Participants ranged in age from 21 to 37 years old, with a mean age of 24 years old. A total of four focus groups were conducted over three months in 2021. The average time of the focus groups was 76 min.

Three major themes were revealed by thematic analysis. The first theme centered on student assertions that deprescribing is vitally important for pharmacists, patients, and allied health professionals. The second theme encompassed perceptions that there are significant barriers to deprescribing. The third theme consisted of the education recommendations, including didactic, clinical, and simulation education, surrounding deprescribing. These themes spotlight the importance of deprescribing, outline the barriers to deprescribing for pharmacists, and offer educational solutions to overcome barriers and improve patient and population health by deprescribing.

**Theme 1: The Importance of Deprescribing: "It's probably one of the forefront problems for pharmacies these days".**

The student pharmacists repeatedly called attention to the importance of deprescribing for patient health and for their future careers.

Student 2 (ST2) asserted that pharmacists play a central role in deprescribing, as they cannot rely on other healthcare professionals to deprescribe:

*"I can't count on another profession to, you know, catch the gaps in a patient's medication history."* (ST2, FG1, Male)

ST3, from another FG echoed a similar sentiment:

*"I totally agree with everyone . . . I think what ST4 said about having patients advocate for themselves, I think something really important is for us, as pharmacists, to advocate for ourselves too. Like when you're at the window, counseling, like reminding your patients to keep that relationship . . . I am someone you can come to when you have these issues."* (ST3, FG2, Female)

ST3 argued that pharmacists are crucial members of the healthcare team, especially in terms of deprescribing.

ST5 also highlighted that deprescribing is particularly important for women's health:

*"One thing that I think, too, that is overlooked is like women's health and like birth control and stuff and all the side effects . . . a lot of women just feel unheard when it comes to stuff like that, and it could really help a lot of people out if they were listened to and taken off that medication."* (ST5, FG2, Female)

In sum, ST5 indicated that deprescribing and deprescribing education could help to ameliorate the healthcare disparities faced by women

Polypharmacy was a key concern for participants, who linked the prevalence of polypharmacy to the importance of deprescribing. ST1 stated:

*"I think deprescribing is absolutely important because, you know, as much as pharmacists like medications, we don't want our patients to be on a bunch of medications because that just causes more issues for us and more issues for them long-term. So, limiting the amount of medications that they have to be on is absolutely amazing, and it's something that we all strive for in the long run just because of its importance and how polypharmacy causes greater issues."* (ST1, FG2, Male)

ST2 agreed and called attention to the unique role pharmacists play in the deprescribing process:

*"I just wanted to echo ST5 there. That's 100% true. Like you have a patient that so many different prescribers and specialists, but you're the one that has the whole list of everything they're taking, right? So, yeah, I mean, I agree. You know, same thing with what ST1 was saying, like we're the drug experts, we have that knowledge."* (ST2, FG2, Male)

In another FG, ST5 again noted that deprescribing is vitally important to reducing polypharmacy:

*"I think deprescribing is extremely important for adherence purposes . . . It could probably upgrade their quality of life by deprescribing and getting down to a smaller amount of medications."* (ST5, FG3, Male)

In summation, ST5 linked deprescribing to polypharmacy and medication adherence. Finally, ST1 wrapped up their focus group's discussion surrounding the importance of deprescribing by emphasizing pharmacist's role in improving patient health:

*"I would agree with everyone, [deprescribing] definitely should be incorporated more into the curriculum, especially at an earlier stage . . . we're not just trying to give people medication, we're trying to benefit their health, so I think it's definitely something that we should learn more about."* (ST4, FG1, Male)

The student pharmacists considered deprescribing to be a crucial part of their role as future pharmacists and identified deprescribing as a potential solution to polypharmacy, an improved medication adherence, and a help to mitigating the healthcare disparities faced by women.

**Theme 2: Barriers to Deprescribing: "Another barrier would be . . . ".**

The student pharmacists commented that the biggest barriers to successful deprescribing are due to prescriber and patient resistance to deprescribing and trust gaps between pharmacists and prescribers and between pharmacists and patients.

ST4 asserted that patients trust physicians more than pharmacists and that pharmacists need to focus on building trust with their patients:

*"Establishing that connection with the patient, I think that's a very big thing that we have to overcome as pharmacists. Like [my fellow focus group participants] said previously, they kind of do trust the doctor more, that's definitely true for a majority of people."* (ST4, FG1, Male)

ST1 echoed that concern:

*"Usually, I have patients who are always like, oh, I don't want to take all these meds, but then they'll take just whatever the doctor gives them. So that's a hurdle that I've definitely ran into in terms of trying to deprescribe."* (ST1, FG2, Male)

In summary, ST1 argued that the trust disparity between patients and physicians and between patients and pharmacists is a major barrier to initiating deprescribing.

ST10 spoke about an unsuccessful attempt to deprescribe a very large dose of an antipsychotic that was driven by the fear of the prescriber on the healthcare team:

*"So, instead of wanting to take her off, they wanted to leave her on [unnecessary medication] just out of fear."* (ST10, FG4, Female)

ST5 provided further examples of unsuccessful attempts to deprescribe related to the trust gap:

*"I've seen the most resistance in the hospital setting is with antibiotics that are no longer—gotten unrecommended, I've seen quite a bit of resistance from doctors wanting to discontinue those. I have also seen quite a bit of a resistance in geriatrics with patients who are on like two benzos, two benzodiazepines, and they don't want to stop one. You know, they've been on them for years, they see the benefit, and they're very resistant to anything we have to say about it."* (ST5, FG4, Female)

In regard to hesitancy about approaching prescribers about deprescribing, ST3 reiterated this common theme:

*"I think just reemphasizing the fear of rejection or just thinking it may be a waste of time, if you don't think they're going to accept your recommendation."* (ST3, FG1, Female)

The student pharmacists did remain optimistic that they could successfully deprescribe by focusing on establishing trust with physicians, as indicated by ST4:

*"I agree with what ST3 said, honestly. There are going to be a lot of times in our lives where we're going to get rejected or maybe even yelled at over . . . so you just accept that . . . Like, if you see something that you think is a valuable opinion and could definitely benefit the patient, I think you should go for it and do it, even if you think you're going to get in trouble– er, not in trouble, but yelled at. There's that percentage chance that you might actually make a difference in someone's life."* (ST4, FG1, Male)

In sum, ST1 again called attention to the vital role that pharmacists play in the healthcare team in pursuit of improving patient health.

In the FGs, the student pharmacists repeatedly linked prescriber and patient resistance to deprescribing to trust gaps between pharmacists and prescribers and between pharmacists and patients. Ultimately, the student pharmacists argued that pharmacists have to overcome these barriers and build trusting relationships with prescribers and patients in order to successfully deprescribe.

**Theme 3: Education Recommendations: "[Deprescribing] needs to be taught soon, you know, at the start of the pharmacy curriculum rather than at the end"**

The student pharmacists were verbose when offering suggestions for improving deprescribing education, which included deprescribing simulations, integrating deprescribing throughout the didactic curriculum, and emphasizing deprescribing during clinical experiences. Table 1 provides an overview of the educational recommendations provided by the student pharmacists in the focus group interviews.

**Table 1.** Deprescribing educational recommendations.

| Simulation | Didactics | Clinical Experiences |
|---|---|---|
| **Simulated conversations with prescribers and patients:** "One thing I think that would make me feel more comfortable with doing [deprescribing] is actually having practice with it within school. I guess we could some on like our rotations, but just getting like the basic grounds of how to build up that conversation you have with people, both patients and prescribers alike." (ST4, FG1, M) | **Curriculum integration coupled with continuing education:** "I think integrating it into the curriculum in the pharmacy school would be helpful and make those students more comfortable as they go through it, but I also feel like things for pharmacists already out of school, so like maybe continuing education on deprescribing, any like seminars and just giving them more exposure as they continue." (ST1, FG1, M) | **Make deprescribing education a routine part of rotations:** "Adding deprescribing to . . . an ambulatory care or hospital rotation . . . I think that would be a great way to incorporate it so that at least the preceptors are like, oh, yeah, I can mention deprescribing in whatever form, whether it's a patient case or a topic discussion . . . I think that's one way to slip in the exposure during rotations." (ST3, FG2, F) |
| **Simulated patient deprescribing conversations:** "Having the opportunity to at least practice with a simulated patient I think would be beneficial." (ST3, FG1, F) | **Direct instruction on titration:** "I think there could be more emphasis on . . . how to deprescribe . . . with like general class of medications, like, okay, someone is taking this, you're going to want to stop them over this period of time." (ST4, FG1) <br> **Titration fundamentals:** "Just more concrete instruction as to how to deprescribe." (ST1, FG3, M) | **A rotation focused primarily on deprescribing:** "It could be like a rotation, like maybe like half of a rotation, if a preceptor wants to be like, hey, for two weeks, I'll have them doing nothing but deprescribe, and then they could get comfortable with talking to physicians and knowing how all of it works, what won't work and stuff like that." (ST5, FG2, F) |
| **Motivational interviewing:** "Motivating them to make lifestyle modifications to be more adhering to their medication . . . use the motivational interviewing strategies to the deprescribing, just motivating them and like trying to get them to understand why we're trying to talk to them about deprescribing whatever medication it is." (ST1, FG1, M) | **Go beyond instruction on prescribing:** "During clinical trials, they always focused on how effective the medication is when patients are on the therapy, but they never talk about deprescribing it in the clinical trials, so it's kind of like systematic, you know, in the industry where it's like, oh, if the research companies don't talk about deprescribing, then the pharmacists aren't going to talk about deprescribing, then the schools aren't going to talk about it." (ST2, FG1, M) | **Preceptor support is key:** "I feel very hesitant in initiating these types of conversations to anyone other than my own preceptor . . . so, personally, for me, I think that like being in that practice and getting experience of being backed up by a preceptor is what really initiates that conversation and really gives you the confidence to talk to your– talk to the provider or the patient directly as well" (ST3, FG2, F) |
| **End of life care:** "I think a big area where deprescribing happens might be a transition to end-of-life care. So that's a very sensitive conversation to have with the patient or patient's family even. So I think more scenarios based on that kind of realm would definitely be helpful." (ST4, FG1, M) | **Instruct on common cases for deprescribing:** "we're taught mostly on the basis of this medication treats this and this medication is for this, but we're not really taught the opposite of like you would not use this in this, or like this azithromycin is not for this, so if you see this, look out for that." (ST3, FG1, M) | **Constant reminders of the importance of deprescribing:** "Having the preceptor . . . instill [deprescribing] in you and having you keep that in the back of your head that that's something you should be looking out for as you progress your career . . . would be a good thing." (ST2, FG2, M) |

**Table 1.** *Cont.*

| Simulation | Didactics | Clinical Experiences |
|---|---|---|
| **Simulation where student pharmacists are tasked with deprescribing:**<br>"I could imagine a similar scenario where, you know, we take a patient's case . . . and we're given the task of deprescribing it." (ST2, FG1, M) | **Identifying red flags:**<br>"As far as adding anything to the curriculum . . . really knowing when we see a medication that doesn't have a use and like knowing how to handle that and seeing it as a red flag would be useful as well." (ST3, FG1, F) | |
| **Hard conversations:**<br>"There really wasn't a lot of these like hard conversations, though . . . I really think that being more comfortable with those kinds of conversations just comes down to . . . having them more often." (ST4, FG1, M) | **Deprescribing lectures:**<br>"I think starting with the lectures probably is just going to be the best way to do it." (ST1, FG2, M) | |
| **Using medication history to deprescribe:**<br>"I don't think it would be too hard to do a simulation where we're given a medication history or something and we have to basically figure out how to talk to the patient about stopping the medication if it's not necessary." (ST3, FG1, F) | **Integrate deprescribing throughout instruction:**<br>"Slip in like deprescribing info for each disease state or drug that you're going over." (ST2, FG2, M) | |

The student pharmacists suggested integrating a variety of deprescribing simulations throughout pharmacy school. ST3 suggested integrating deprescribing simulations into the pharmacy curriculum:

*"I don't think it would be too hard to do a simulation where we're given a medication history or something and we have to basically figure out how to talk to the patient about stopping the medication if it's not necessary."* (ST3, FG1, Female)

ST2 agreed stating:

*"Similar to what ST3 was saying . . . I could imagine a similar scenario where, you know, we take a patient's case . . . and we're given the task of deprescribing it."* (ST2, FG1, Male)

ST2 saw a need for simulations that include deprescribing discussions with prescribers:

*"I guess we haven't really had like—I know it would be hard to do for the school, but like actual one-on-one talking with a person that's, I guess, supposed to be a prescriber."* (ST2, FG3, Male)

ST4 reflected on the professional communication simulations they experienced and called for frequent practice with the hard conversations related to deprescribing:

*"We definitely did get a lot of practice of like good professional communication . . . There really wasn't a lot of these like hard conversations, though . . . I really think that being more comfortable with those kinds of conversations just comes down to . . . having them more often."* (ST4, FG1, Male)

ST3 expanded on the simulation discussion and suggested adding deprescribing education throughout the curriculum:

*"I agree with what ST1 said . . . having the opportunity to at least practice with a simulated patient I think would be beneficial. As far as adding anything to the curriculum, just like for our classes, really knowing when we see a medication that doesn't have a use and like knowing how to handle that and seeing it as a red flag would be useful as well."* (ST3, FG1, Female)

In addition to deprescribing simulations, the student pharmacists called for more didactic deprescribing instruction. ST2 suggested adding deprescribing to lectures and asked faculty to *"slip in like deprescribing info for each disease state or drug that you're going over."* (ST2, FG2)

ST1 concurred:

*"I agree with ST2 100%. I think starting with the lectures probably is just going to be the best way to do it."* (ST1, FG2, Male)

ST4 offered a similar suggestion:

*"I think there could be more emphasis on like how to deprescribe . . . I wish they would have went more with like general class of medications, like, okay, someone is taking this, you're going to want to stop them over this period of time."* (ST4, FG1, Male)

ST1 called for integrating deprescribing into the curriculum and suggested offering continuing education focused on deprescribing:

*"I think integrating it into the curriculum in the pharmacy school would be helpful and make those students more comfortable as they go through it, but I also feel like things for pharmacists already out of school, so like maybe continuing education on deprescribing, any like seminars and just giving them more exposure as they continue."* (ST1, FG1, Male)

In addition to simulation and didactic instruction, the student pharmacists called for preceptors to place a greater emphasis on deprescribing during clinical experiences. ST3 suggested adding objectives to clinical rotations with preceptors to increase exposure to deprescribing:

*"So kind of going off that . . . I think adding deprescribing to maybe like an ambulatory care or hospital rotation . . . I think that would be a great way to incorporate it so that at least the preceptors are like, oh, yeah, I can mention deprescribing in whatever form, whether it's a patient case or a topic discussion, something like that. I think that's one way to slip in the exposure during rotations."* (ST3, FG2, Female)

ST5 concurred:

*"It could be like a rotation, like maybe like half of a rotation, if a preceptor wants to be like, hey, for two weeks, I'll have them doing nothing but deprescribe, and then they could get comfortable with talking to physicians and knowing how all of it works, what won't work and stuff like that."* (ST5, FG2, Female)

ST2 echoed the sentiment:

*"I agree . . . I think [deprescribing should be looked at more . . . So, like having the preceptor sort of, you know, instill that in you and having you keep that in the back of your head that that's something you should be looking out for as you progress your career or whatever I think would be a good thing."* (ST2, FG2, Male)

ST3 shared their experience with preceptors discussing deprescribing and argued that preceptor education on deprescribing was paramount to building the confidence to deprescribe:

*"I totally echo what everyone had said, but like as a student, I feel very hesitant in initiating these types of conversations to anyone other than my own preceptor . . . so, personally, for me, I think that like being in that practice and getting experience of being backed up by a preceptor is what really initiates that conversation and really gives you the confidence to talk to your—talk to the provider or the patient directly as well."* (ST3, FG2, Female)

The deprescribing education recommendations centered around simulation, didactics, and clinical rotations. The student pharmacists proposed integrating deprescribing into the existing pharmacy curriculum, adding deprescribing-focused simulations, and providing guidance to preceptors surrounding deprescribing education during clinical experiences.

## 4. Discussion

The three emergent themes offer insight into the importance of deprescribing for pharmacists and the barriers faced when initiating deprescribing and offer recommendations for improving pharmacy education on deprescribing. With the increased prevalence of polypharmacy in the US, understanding the barriers pharmacists face when deprescribing and recommendations to enhance deprescribing education in the pharmacy curriculum is needed [4].

Focus groups were conducted to better understand student pharmacists' perspectives on deprescribing. During these focus groups, the participants were invited to share their experiences with and thoughts about deprescribing. The themes established from the student pharmacists' responses during these interviews provide a justification for an increased emphasis on deprescribing education, call for contemplation on the barriers to deprescribing faced by pharmacists, and offer a roadmap for improving deprescribing education for student pharmacists.

The student pharmacists identified deprescribing and polypharmacy as pressing, interrelated issues facing pharmacists today. Throughout the interviews, the student pharmacists repeatedly emphasized the importance of deprescribing to improve patient health [20]. The participants viewed deprescribing as a key function of their role as pharmacists. They called attention to the fact that pharmacists play a key role in deprescribing as they often have the most comprehensive view of their patient's medication regimen, which is supported by Reeder and Mutnick, who "found that pharmacist-obtained medication histories resulted in less discrepancies and more-thorough medication histories than did physician-obtained medication histories" [21]. The student pharmacists suggested that

deprescribing could be a possible solution to polypharmacy, an aid to increase medication adherence, and a step towards combating the healthcare challenges faced by women [22,23]. The assertion that deprescribing can reduce polypharmacy is supported by Sun et al., who stated: "The reduction of polypharmacy prevalence rates, along with a decrease in the associated [adverse drug reactions] can be accomplished through the process of deprescribing" [13].

While the student pharmacists argued for the importance of deprescribing, they also highlighted that initiating deprescribing faces significant barriers. During focus groups, the student pharmacists shared their unsuccessful deprescribing experiences, which they attributed to patient and prescriber resistance to deprescribing. These concerns align with Reeve's work on deprescribing tools [6]. Reeve stated that "tools which highlight inappropriate medications may not be effective at increasing deprescribing as time is needed to assess the appropriateness in the individual, discuss it with the patient and/or caregivers and plan the deprescribing process" [6]. Ultimately, the student pharmacists linked their failed deprescribing experiences to the strong trust patients place in prescribers and their own limited relationships with patients. The participants called attention to the strong trust bond between prescribers and patients that often dwarfed their own trust bond with patients. They also spoke to the need for developing strong, trusting relationships between pharmacists and patients. To overcome patient and prescriber resistance to deprescribing, the student pharmacists suggested focusing on building strong relationships with patients and on building trust between pharmacists and prescribers. Emphasizing the importance of strengthening relationships between pharmacists and patients may be promising for deprescribing and limiting polypharmacy, as Waszyk-Nowacyzk et al. found: of the patients they surveyed at a single-center in a large Polish city, "79.4% of patients would like to benefit from medicines use reviews provided by a pharmacist" [24]. Additionally, pharmacists, in Poland, interviewed by Łucja Zielińska-Tomczak et al. on the topic of interprofessional collaboration "indicated that the younger generation of physicians seems more cooperative than older doctors", while the physicians who were interviewed in Poland "supported these views and suggested a positive effect of informal relationships with pharmacists on doctors' openness towards collaboration" [25].

To achieve the benefits of deprescribing, such as improving patient health, and to overcome the barriers to deprescribing, the participants proposed adopting new educational practices surrounding deprescribing simulation, didactics, and clinical experiences. Other researchers also concluded, based on their survey of trainees in pharmacy medicine and nursing, that "alterations to the current curricular content may be warranted to address lack of preparedness to deprescribe in clinical practice" [9].

The pharmacy students called for deprescribing simulations with prescribers and with patients so they could practice those difficult conversations and receive feedback and advice. This aligns with Palaganas, Epps, and Raemer, who argued that "given the increasing adoption of experiential learning and team-based learning, [healthcare simulation] has become a preferred vehicle for [interprofessional education]" [26]. Participants asked for deprescribing to be integrated within the didactic curriculum whenever discussing disease states or drugs. This desire for additional didactic instruction on deprescribing tracks with a survey of stunt pharmacists by Clark et al., who found that "less than half of students felt that their didactic training adequately prepared them for deprescribing in the clinical setting" [10]. In addition, the student pharmacists recommended that pharmacy schools give preceptors learning objectives based on deprescribing and to make deprescribing part of the clinical curriculum. These recommendations are supported by Raiman-Wilms et al., who argued that "Teaching health professional learners how to apply evidence into clinical shared decision-making is important in the teaching of prescribing and deprescribing" [12].

Deprescribing is vitally important to reducing polypharmacy and improving patient health [5,7,13]. While there are significant barriers to deprescribing, the educational recommendations offered here are a meaningful step towards integrating deprescribing into the everyday practice of pharmacists and student pharmacists alike.

*4.1. Strengths, Limitations, and Future Studies*

This study offers a broad view on student pharmacists' perceptions of deprescribing, as it was compromised of a heterogeneous sample size of student pharmacists from geographical locations throughout the US (East, South, and West). The student pharmacists sampled were predominately third- and fourth-year students. As with any study of this nature, it is possible that only those with an interest in this topic consented to participate in the study, which may be a source of sampling bias. While third- and fourth-year students can provide a comprehensive take on deprescribing education throughout the pharmacy curriculum, future research may focus on the recruitment of first- and second-year students pharmacists to better understand their unique perspectives on deprescribing.

The use of TPB to guide the qualitative study design facilitated the recording of a heterogeneous sample of student pharmacist voices. Additionally, the use of video conferencing software aided in the recruitment of student pharmacists from across the US to offer their experiences with and perspectives on deprescribing. The sample characteristics and recruitment methods employed may, however, limit the generalizability of the findings. The findings from this study call for more longitudinal research on deprescribing and the collection of perspectives from across the healthcare education system.

*4.2. Conclusions and Future Studies*

This study builds on previous work that discussed student pharmacists' limited knowledge about deprescribing and their belief that deprescribing education is necessary [8]. This study expands upon that previous work by calling attention to the importance of deprescribing in pharmacy today and providing educational suggestions to overcome barriers to deprescribing through three emerging themes: (1) deprescribing improves patient care and is a necessary skill for pharmacists; (2) there are significant barriers to deprescribing; (3) barriers to deprescribing can be overcome by providing additional educational opportunities. With the recognized benefits of deprescribing for patients, the student pharmacists see the need to become proficient in deprescribing, yet there are significant barriers that pharmacists continue to face when deprescribing. The student pharmacists believe that the integration of deprescribing content in clinical, didactic, and simulation education will help them to overcome the barriers to deprescribing and to positively impact patient care. Colleges of pharmacy should consider emphasizing the importance of deprescribing, creating programs to assist future pharmacists in addressing barriers to deprescribing, and adopting the educational strategies suggested to improve deprescribing education.

**Author Contributions:** Conceptualization, S.P.S., R.E.B. and A.C.; methodology, A.C. and D.S.; software, A.C., D.S. and D.R.A.; validation, A.C., D.S. and D.R.A.; formal analysis, A.C., D.S. and D.R.A.; investigation, A.C., D.S. and D.R.A.; resources, A.C., S.P.S., R.E.B., D.S. and D.R.A.; writing—original draft preparation, A.C., S.P.S., D.S. and D.R.A.; writing—review and editing, A.C., D.R.A., R.E.B., S.P.S. and D.S.; project administration, A.C., D.S., S.P.S. and D.R.A. All authors have read and agreed to the published version of the manuscript.

**Funding:** This research received no external funding.

**Institutional Review Board Statement:** The study was conducted according to the guidelines of the Declaration of Helsinki and was approved by the Institutional Review Board (or Ethics Committee) of the University of Tennessee Health Science Center (IRB# 21-08234-XM), University of Arizona (IRB# 2021-015-PHPR), and University of New England (IRB# 0821).

**Informed Consent Statement:** Consent was received from all the participants prior to conducting the study.

**Data Availability Statement:** Data sharing not applicable to this article.

**Conflicts of Interest:** All authors declare no conflict of interest.

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
