# Peer review of "Teaching Deprescribing and Combating Polypharmacy in the Pharmacy Curriculum: Educational Recommendations from Thematic Analysis of Focus Groups"

_clinpract, doi:10.3390/clinpract13020040_

Round 1

Reviewer 1 Report

The study in question investigates the views of student pharmacists on the importance of deprescribing, their perceived barriers to it, and their attitudes and recommendations on incorporating deprescribing education into pharmacy program curricula. The authors conducted interviews with students recruited from three different universities using a semi-structured approach and qualitative thematic analysis.

Deprescribing is an essential role for pharmacists, and student pharmacists should possess the necessary knowledge and skills to deprescribe medications appropriately, especially in cases where polypharmacy is a concern. The current study comes with major issues that need to be addressed.

1.                 The Methods section is lacking important details. The authors should provide enough information to allow the reader to evaluate the usefulness of the findings. When relevant, the authors should include the gender of participants, the recruitment period, the number of focus groups and students per group, the method of assigning students to groups, the number of sessions per group, the timeframe of these sessions, the length of each session, the main starting questions asked in each group, and if the same questions were asked to all groups.

2.                   A discussion should be included, possibly in the limitations section, addressing the low acceptance rate of invitations (26 out of 1366) and its potential impact on the study's external validity.

3.                   The authors should address the limitations of the study in terms of the representativeness of the sample and the generalizability of the results.

4.                   The authors should address the limitation of subjectivity in the study, which may reflect the students' views rather than the actual content of the pharmacy curriculum. Have the authors considered examining the curriculum itself instead of relying solely on student views? Including pharmacy teachers in the focus groups to compare student perceptions with actual teachings could be useful. If this was not done, the authors may introduce it as a potential future direction for the current study.

5.                   Line 53: The sentence "Another study found that student pharmacists felt more prepared to identify inappropriate medications than medical students (REF#9)" seems to contradict the earlier sentence "student pharmacists are often unprepared to recommend or implement deprescribing in practice (REF#8-10)". The same reference (REF#9) is cited for both statements. This discrepancy needs to be addressed by the authors to avoid any confusion to the reader. It is possible that the study may have found conflicting results or that the authors may have misinterpreted the findings. Clarification is necessary.

6.                   Line 82: “A semi-structured fgtrategy”. Replace with “A semi-structured strategy”.

7.                   Line 104: “Participants from three colleges of pharmacy in the US, totaling 1,366 student pharmacists, were invited to participate in this study”. The study had 26 participants. Those that were approached for participating can not be called “participants”. Please modify the statement accordingly.

Author Response

The study in question investigates the views of student pharmacists on the importance of deprescribing, their perceived barriers to it, and their attitudes and recommendations on incorporating deprescribing education into pharmacy program curricula. The authors conducted interviews with students recruited from three different universities using a semi-structured approach and qualitative thematic analysis.

Deprescribing is an essential role for pharmacists, and student pharmacists should possess the necessary knowledge and skills to deprescribe medications appropriately, especially in cases where polypharmacy is a concern. The current study comes with major issues that need to be addressed.

  1.                The Methods section is lacking important details. The authors should provide enough information to allow the reader to evaluate the usefulness of the findings. When relevant, the authors should include the gender of participants, the recruitment period, the number of focus groups and students per group, the method of assigning students to groups, the number of sessions per group, the timeframe of these sessions, the length of each session, the main starting questions asked in each group, and if the same questions were asked to all groups.

Response: Thank you for your valuable feedback. We included additional information in the methods regarding the recruitment period the total number of focus groups. We also added information about the time frame and the gender of the participants.

  1. A discussion should be included, possibly in the limitations section, addressing the low acceptance rate of invitations (26 out of 1366) and its potential impact on the study's external validity.

Response: Thank you for this critical feedback. A sample size of 26 is sufficient for a qualitative study since we were looking to reach thematic saturation. For quantitative research, we agree that large sample size and a reasonable response rate are essential for external validity. Instead, in qualitative research, data saturation is critical to address. 

  1. The authors should address the limitations of the study in terms of the representativeness of the sample and the generalizability of the results.

Response: We have added as a limitation that the students who participated in this study may have been more interested in the topic, thus potentially biasing the sample. However, this would be the case whenever participants are asked to participate in a study. In addition, please see our response above about generalizability.

  1. The authors should address the limitation of subjectivity in the study, which may reflect the students' views rather than the actual content of the pharmacy curriculum. Have the authors considered examining the curriculum itself instead of relying solely on student views? Including pharmacy teachers in the focus groups to compare student perceptions with actual teachings could be useful. If this was not done, the authors may introduce it as a potential future direction for the current study.

Response: Thank you for this suggestion to examine the curriculum; however, this is not the scope of this study. Furthermore, the American Association of Colleges of Pharmacy (AACP) has very robust guidelines regarding the pharmacy curriculum. Thus, with this aim in mind, we decided to inform our respective colleges and let them know the students' perspectives.

  1. Line 53: The sentence "Another study found that student pharmacists felt more prepared to identify inappropriate medications than medical students (REF#9)" seems to contradict the earlier sentence "student pharmacists are often unprepared to recommend or implement deprescribing in practice (REF#8-10)". The same reference (REF#9) is cited for both statements. This discrepancy needs to be addressed by the authors to avoid any confusion to the reader. It is possible that the study may have found conflicting results or that the authors may have misinterpreted the findings. Clarification is necessary.

Response: Thank you for this clarification. The manuscript addressed it.

  1. Line 82: “A semi-structured fgtrategy”. Replace with “A semi-structured strategy”.

Response: Thank you for pointing out this typo. It was addressed.

  1. Line 104: “Participants from three colleges of pharmacy in the US, totaling 1,366 student pharmacists, were invited to participate in this study”. The study had 26 participants. Those that were approached for participating can not be called “participants”. Please modify the statement accordingly.

Response: Please see our revised text to clarify this point.

Reviewer 2 Report

Dear Authors,

Thank you for the opportunity to review the paper “The Importance of, Barriers to, and Educational Recommendations for Teaching Deprescribing in the Pharmacy Curriculum: Findings from Thematic Analysis of Focus Groups”. Although I found it very informative and interesting, I have several suggestions, which I hope, after their implementation, could improve the paper.

1) lines 66-69: the aim of the study – maybe it would be better to change it into something like “… perceptions of the implementation of the place of deprescribing in the pharmacy curriculum.”

2) line 82 in Methods – the word “fgtrategy” – is it correct, or did you mean FG strategy?

3) line 85 – This is one of my two major reservations. You mention the use of the Theory of Planned Behavior. However, it is not evidenced anywhere else in your paper. What exactly is the behavior under question? – Did you follow the TACT principle (Target, Action, Context and Time)? What part of your results is relevant to which of the TPB’s variables – attitudes, subjective norms and perceived behavioral control? What are the behavioral intentions of your participants? These are all important questions that should be mentioned (at least in the discussion part) if TPB is really to be applied. There is a dedicated manual to research using TPB – maybe you will find it helpful: Francis, J., Eccles, M. P., Johnston, M., Walker, A. E., Grimshaw, J. M., Foy, R., Kaner, E. F. S., Smith, L. and Bonetti, D. (2004). Constructing questionnaires based on the theory of planned behaviour: A manual for health services researchers. Newcastle upon Tyne, UK: Centre for Health Services Research, University of Newcastle upon Tyne.

4) The questions asked (either a list or an outline of topics) should be provided.

5) Did you seek approval from an appropriate Ethics Committee/Regulatory Body? Or was it not necessary under US jurisdiction?

6) My second major reservation revolves around the quality of the discussion. In my opinion, it mostly involves a repeat of the results with a very limited actual discussion with relevant literature (4 papers are cited if I counted correctly). Given the abundance of available literature, this is something that must be amended. Firstly, as I have written above, the reference to TPB should be introduced – I think that it could constitute a basis or core of your discussion. Secondly, more literature on the topic could be provided (not necessarily only from the US – it could increase the international relevance of your paper).

To help you start – in lines 336-348 – you mention patient and prescriber resistance to deprescribing. However, it may not necessarily always be the case, and now the paper presents only one side of the story. For example, in a recent study, 79.4% of patients said that they would like to benefit from medicines use review conducted by pharmacists - https://www.mdpi.com/1660-4601/20/2/945

On the other hand, the relationship between prescribers and pharmacists was described in detail in another paper (including the topic of pharmacists’ involvement in patient pharmacotherapy with an emphasis on the pharmacist’s role as an expert on drugs) which also used the Theory of planned behavior – maybe you will find it useful - https://bmchealthservres.biomedcentral.com/articles/10.1186/s12913-021-06903-5

7) lines 341-345 – please verify if the combinations of prescribers, pharmacists, and patients are correct. Because you mention a trust gap between prescribers and patients, then a strong trust bond between them, and then the need for such a bond. Did you mean pharmacists instead of prescribers in some of them?

Good luck!

Author Response

Thank you for the opportunity to review the paper “The Importance of, Barriers to, and Educational Recommendations for Teaching Deprescribing in the Pharmacy Curriculum: Findings from Thematic Analysis of Focus Groups”. Although I found it very informative and interesting, I have several suggestions, which I hope, after their implementation, could improve the paper.

1) lines 66-69: the aim of the study – maybe it would be better to change it into something like “… perceptions of the implementation of the place of deprescribing in the pharmacy curriculum.”

Response: Thank you for this valuable suggestion. We incorporated in our manuscript that reads much better.

2) line 82 in Methods – the word “fgtrategy” – is it correct, or did you mean FG strategy?

Response: Thank you for pointing out this typo. It was addressed.

3) line 85 – This is one of my two major reservations. You mention the use of the Theory of Planned Behavior. However, it is not evidenced anywhere else in your paper. What exactly is the behavior under question? – Did you follow the TACT principle (Target, Action, Context and Time)? What part of your results is relevant to which of the TPB’s variables – attitudes, subjective norms and perceived behavioral control? What are the behavioral intentions of your participants? These are all important questions that should be mentioned (at least in the discussion part) if TPB is really to be applied. There is a dedicated manual to research using TPB – maybe you will find it helpful: Francis, J., Eccles, M. P., Johnston, M., Walker, A. E., Grimshaw, J. M., Foy, R., Kaner, E. F. S., Smith, L. and Bonetti, D. (2004). Constructing questionnaires based on the theory of planned behaviour: A manual for health services researchers. Newcastle upon Tyne, UK: Centre for Health Services Research, University of Newcastle upon Tyne.

Response: We truly appreciate your valuable suggestion—our previous manuscript presented in-depth information on how the TPB was used in constructing the Focus Group guide. So again, thank you for these recommendations on how to construct the guide; however, our study was qualitative, and more information regarding the division of attitudes, subjective norms, and perceived behavioral control were presented in our previous manuscript.

4) The questions asked (either a list or an outline of topics) should be provided.

Response: Thank you for this suggestion. The previous manuscript provided additional information about the interview guide.

5) Did you seek approval from an appropriate Ethics Committee/Regulatory Body? Or was it not necessary under US jurisdiction?

Thank you for suggestion. In the U.S. we use the Institutional Review Board (IRB). We obtained approval for each institution. We have included text at the start of the methods to state the IRB approvals: “including the University of Tennessee Health Science Center (UTHSC, IRB# 21-08234-XM), University of Arizona (IRB# 2021-015-PHPR), and University of New England (UNE, IRB# 0821)”. This information is also provided towards the end of the manuscript, in the “Institutional Review Board Statement” section.

6) My second major reservation revolves around the quality of the discussion. In my opinion, it mostly involves a repeat of the results with a very limited actual discussion with relevant literature (4 papers are cited if I counted correctly). Given the abundance of available literature, this is something that must be amended. Firstly, as I have written above, the reference to TPB should be introduced – I think that it could constitute a basis or core of your discussion. Secondly, more literature on the topic could be provided (not necessarily only from the US – it could increase the international relevance of your paper).

Response: Thank you for your feedback and for sharing those sources. They have been integrated into the discussion section alongside additional contextualization and inclusion of relevant literature.

To help you start – in lines 336-348 – you mention patient and prescriber resistance to deprescribing. However, it may not necessarily always be the case, and now the paper presents only one side of the story. For example, in a recent study, 79.4% of patients said that they would like to benefit from medicines use review conducted by pharmacists - https://www.mdpi.com/1660-4601/20/2/945

Response: We are so grateful for these valuable resources. We integrated them and the manuscript is much stronger. Thank you!

Round 2

Reviewer 1 Report

I would like to thank the authors for revising the manuscript. I would love to endorse the paper for publication. However, while the authors have addressed the comments raised, it appears that not all of the changes mentioned in the reply have been reflected in the tracked manuscript. Did the authors track all modifications? It would be helpful if the authors could either include the modifications they have made in their response, indicate the lines in the revised manuscript where changes have been made, or appropriately track all changes made to the previous version of the manuscript.

Author Response

I would like to thank the authors for revising the manuscript. I would love to endorse the paper for publication. However, while the authors have addressed the comments raised, it appears that not all of the changes mentioned in the reply have been reflected in the tracked manuscript. Did the authors track all modifications? It would be helpful if the authors could either include the modifications they have made in their response, indicate the lines in the revised manuscript where changes have been made, or appropriately track all changes made to the previous version of the manuscript.

Response: We apologize for the inconvenience.

We added the sentences and the information to each suggestion.

  1.                The Methods section is lacking important details. The authors should provide enough information to allow the reader to evaluate the usefulness of the findings. When relevant, the authors should include the gender of participants, the recruitment period, the number of focus groups and students per group, the method of assigning students to groups, the number of sessions per group, the timeframe of these sessions, the length of each session, the main starting questions asked in each group, and if the same questions were asked to all groups.

Response: Thank you for your valuable feedback. We included additional information in the methods regarding the recruitment period the total number of focus groups. We also added information about the time frame and the gender of the participants.

“The recruitment occurred simultaneously over three months in the Fall of 2022 semester in all three Colleges of Pharmacy. The interested student pharmacists signed up on a list, and AC contacted them to find out the preferred date and time they would like to participate in the study.”

  1. A discussion should be included, possibly in the limitations section, addressing the low acceptance rate of invitations (26 out of 1366) and its potential impact on the study's external validity.

Response: Thank you for this critical feedback. A sample size of 26 is sufficient for a qualitative study since we were looking to reach thematic saturation. For quantitative research, we agree that large sample size and a reasonable response rate are essential for external validity. Instead, in qualitative research, data saturation is critical to address. 

“Springer et al., 2022 provided additional information regarding the conceptualization of this study and how the TPB guided the development of the questions.3”

“The questions were divided into three topics, and this manuscript focused on deprescribing education.8

  1. The authors should address the limitations of the study in terms of the representativeness of the sample and the generalizability of the results.

Response: We have added as a limitation that the students who participated in this study may have been more interested in the topic, thus potentially biasing the sample. However, this would be the case whenever participants are asked to participate in a study. In addition, please see our response above about generalizability.

“Student pharmacists from three colleges of pharmacy in the US were invited to participate in this study. The total number of eligible student pharmacists was 1,366 (UNE, N = 158; University of Arizona, N = 526; UTHSC, N = 682). Of these, 26 student pharmacists consented to participate in four FG discussions. Most participants (N = 16) were enrolled in their fourth year, while seven were enrolled in the third year, and three were enrolled in their second year. Participants ranged in age from 21 to 37 years old, with a mean age of 24 years old. A total of four FGs were conducted over three months in 2021. The average time of the FG was 76 minutes.”

  1. The authors should address the limitation of subjectivity in the study, which may reflect the students' views rather than the actual content of the pharmacy curriculum. Have the authors considered examining the curriculum itself instead of relying solely on student views? Including pharmacy teachers in the focus groups to compare student perceptions with actual teachings could be useful. If this was not done, the authors may introduce it as a potential future direction for the current study.

Response: Thank you for this suggestion to examine the curriculum; however, this is not the scope of this study. Furthermore, the American Association of Colleges of Pharmacy (AACP) has very robust guidelines regarding the pharmacy curriculum. Thus, with this aim in mind, we decided to inform our respective colleges and let them know the students' perspectives.

“With the increased prevalence of polypharmacy in the US, understanding the barriers pharmacists face when deprescribing and recommendations to enhance deprescribing education in the pharmacy curriculum is needed.4

“Student pharmacists suggested that deprescribing could be a possible solution to polypharmacy, an aid to increase medication adherence, and a step towards combatting healthcare challenges faced by women.22,23

  1. Line 53: The sentence "Another study found that student pharmacists felt more prepared to identify inappropriate medications than medical students (REF#9)" seems to contradict the earlier sentence "student pharmacists are often unprepared to recommend or implement deprescribing in practice (REF#8-10)". The same reference (REF#9) is cited for both statements. This discrepancy needs to be addressed by the authors to avoid any confusion to the reader. It is possible that the study may have found conflicting results or that the authors may have misinterpreted the findings. Clarification is necessary.

To overcome patient and prescriber resistance to deprescribing, student pharmacists suggested focusing on building strong relationships with patients and on building trust between pharmacists and prescribers. Emphasizing the importance of strengthening relationships between pharmacists and patients may be promising for deprescribing and limiting polypharmacy, as Waszyk-Nowacyzk, et al. found that, of the patients they surveyed at a single-center in a large Polish city, “79.4% of patients would like to benefit from medicines use reviews provided by a pharmacist.”24 Additionally, pharmacists, in Poland, interviewed by Łucja Zielińska-Tomczak, et al. on the topic of interprofessional collaboration “indicated that the younger generation of physicians seems more cooperative than older doctors,” while the physicians, in Poland, interviewed “supported these views and suggested a positive effect of informal relationships with pharmacists on doctors’ openness towards collaboration.”25

  1. Line 82: “A semi-structured fgtrategy”. Replace with “A semi-structured strategy”.

Response: Thank you for pointing out this typo. It was addressed.

  1. Line 104: “Participants from three colleges of pharmacy in the US, totaling 1,366 student pharmacists, were invited to participate in this study”. The study had 26 participants. Those that were approached for participating can not be called “participants”. Please modify the statement accordingly.

Response: Please see our revised text to clarify this point.

Of these, 26 student pharmacists consented to participate in four FG discussions. Most participants (N = 16) were enrolled in their fourth year, while seven were enrolled in the third year, and three were enrolled in their second year. Participants ranged in age from 21 to 37 years old, with a mean age of 24 years old. A total of four FGs were conducted over three months in 2021. The average time of the FG was 76 minutes.”

Reviewer 2 Report

The authors have sufficiently acknowledged my previous comments. 

Author Response

Thank you for your time to review our manuscript.

Thank you for the suggestions that strengthen our manuscript.